# Predictive Factors of Toddlers’ Sleep and Parental Stress

**DOI:** 10.3390/ijerph17072494

**Published:** 2020-04-06

**Authors:** Simona De Stasio, Francesca Boldrini, Benedetta Ragni, Simonetta Gentile

**Affiliations:** 1Department of Human Studies, LUMSA University, 00193 Rome, Italy; f.boldrini@lumsa.it (F.B.); b.ragni@lumsa.it (B.R.); 2Unit of Clinical Psychology, Department of Neuroscience and Neurorehabilitation, Bambino Gesù Children’s Hospital, 00165 Rome, Italy; simonetta.gentile@opbg.net

**Keywords:** sleep, toddlers, parental psychosocial functioning

## Abstract

Background: Although most infants consolidate their sleep habits during the first year of life, for many children, sleep is described as disrupted during toddlerhood. Along with individual child variables such as temperamental characteristics, parenting behaviors play a key role in determining children’s sleep–wake patterns. The aims of the current study were to evaluate the relationship among toddlers’ sleep quality, emotion regulation, bedtime routines, parental bedtime involvement, parental perceived social support and stress, and to integrate a novel combination of the aforementioned dimensions into predictive models of toddlers’ sleep quality and parental stress. Methods: One hundred and sixty parents with 2–3-year-old children filled out the following self-report questionnaires: the Parent–Child Sleep Interaction Scale; the Emotion Regulation Checklist; the Social Provisions Scale; and an ad-hoc questionnaire to assess parental involvement in everyday and bedtime care for children. Three multiple regression analyses were conducted by regressing maternal and paternal parenting stress and infant’s quality sleep onto the independent variables described above. Results: Toddlers’ emotion regulation and parental psychosocial functioning were related to parental stress. Toddlers’ night awakenings and the time required by toddlers to fall asleep were related to parental distress. Conclusions: The findings evidenced the bidirectional associations among the studied variables, highlighting the protective role of social support in reducing parenting stress and of paternal bedtime involvement in improving toddlers’ sleep quality.

## 1. Introduction

The establishment of a well-regulated sleep–wake rhythm is a critical and complex developmental process in early infancy [1,2]. Although most infants consolidate their sleep habits during the first year of life, for many children, sleep is described as disrupted during toddlerhood [3]. Commonly, toddlers present difficulties in initiating or maintaining sleep and in returning to bed after night awakenings; furthermore, parents wishing to encourage healthy sleeping behaviors may find it challenging to define appropriate rules and limits [3]. Along with individual child variables such as neurodevelopmental or temperamental characteristics, parenting behaviors appear to play a key role in determining children’s normative sleep–wake patterns [1]. According to Alfano and colleagues [1], a wide range of parenting practices and parent–child interactions can favor or interfere with children’s sleep regulation.

As documented in the literature, the establishment of routines during infancy has positive effects on children’s patterns of biological regulation. According to Mindell et al. [4], customary nightly routines induce a significant improvement in toddlers’ quality of sleep. “Bedtime routines” may be defined as habitual family patterns of behavior, whereby parents engage their child in recurring activities in the same sequence every day before bed. Based on prior research, Tikotzky [5] reported that, in preschoolers, stable routines are associated with shorter sleep latency, reduced night-awakening, less bedtime resistance, and longer night-time sleep duration—all markers of good-quality sleep [4].

With regards to individual child variables, children’s temperamental characteristics, such as a tendency to experience and express negative emotionality with higher intensity and frequency [6], represented risk factors for sleep quality [7]. Emotion regulation is the ability to decrease, maintain, or increase emotional arousal to facilitate engagement with the context [8,9]. Existing literature showed the critical role of caregiver behaviors and the caregiver–child interaction, considering this to be the context in which children acquire their regulatory capacities [10]. Children’s regulation of affective responses and sleep patterns share a common developmental background, with both factors bidirectionally affecting one another [11]. Emotional arousal and difficulty in regulating negative emotionality both influence sleep problems and lead to family processes whereby parents inadvertently maintain a sleep problem by acting to avoid child distress [12,13].

Considering parental behaviors and well-being, recent research on the role of parental psychological functioning in young children’s sleep quality has mainly focused on parental depression as a predictor of sleep disturbances [14]. Hughes et al. [15] observed that mothers of 9-month-old infants with less favorable sleep profiles reported depressive symptoms, poorer self-reported health, and significant levels of parenting stress. De Stasio and colleagues [16] found that mothers and fathers who reported significant levels of parental distress tended to describe their toddlers’ bedtime routines as particularly challenging.

Additionally, it has been shown that parents’ perceived social support is strongly associated with their capacity for adjustment [17]. With respect to sleep, Morrell and Steele [18] reported empirical evidence for an association between disrupted sleep–wake patterns in young children and parental psychosocial distress underpinned by perceptions of poor social support. Conversely, higher levels of support can help parents to establish a positive family environment characterized by stability and cohesion—relevant conditions for regular sleep–wake patterns during infancy [19].

Previous studies have also documented the role of both the daytime and bedtime quality of parenting and parental strategies in predicting sleep disruptions in early childhood [20,21]. Responsive mothers fostered better quality and prolonged nighttime sleep in their infants [22]; by contrast, greater parental involvement, a shorter response latency to infant awakenings and active soothing at bedtime negatively affect children’s quality of sleep (*ibid.*). While the role of mothers in children’s sleep has been described in-depth in the literature, the contribution of fathers has scarcely been investigated to date [14]. Fathers who reported higher marital satisfaction and social support, and lower levels of parenting stress [2], seem to be more involved in caring for their children at bedtime. Paternal involvement and their provision of reliable emotional support to the child [19] positively improve both children’s and mothers’ sleep consolidation [23,24].

From a transactional family dynamics perspective [25], children’s developmental achievements, including sleep consolidation, are conceptualized as influencing and being influenced by multiple subsystems within the larger family system [2]. Research focusing on parenting and infants’ sleep suggests the existence of reciprocal influences between these factors [26]. Children’s sleep problems undermine parents’ sleep patterns and thereby their capacity to regulate emotion [27], which is essential for positive parenting [2].

### Current Study

The aims of the current study were to evaluate the relationship between toddlers’ sleep quality, emotion regulation, bedtime routines, parental bedtime involvement, and parental perceived social support and stress, and to integrate a novel combination of the aforementioned dimensions into predictive models of toddlers’ sleep quality and parental stress. We hypothesized that longer sleep onset latency, greater nocturnal awakenings, children’s emotional lability, and problematic bedtime routines would predict both maternal and paternal parenting stress (H1). On the contrary, we hypothesized that paternal global and bedtime involvement in children’s care would be associated with lower levels of parental stress (H2). Moreover, we expected that longer sleep onset latency, children’s emotional lability, problematic bedtime routines, and greater parenting stress would predict toddlers’ nocturnal awakenings (H3), and that paternal global and bedtime involvement in children’s care would be associated with fewer nocturnal awakenings (H4).

To the best of our knowledge, this is the first study that has examined the role of multiple dimensions of parental psychosocial functioning, integrating all of these variables into a novel combination of psychosocial dimensions which contribute to explaining toddlers’ sleep quality and parenting stress, gathering data from both mothers and fathers.

## 2. Materials and Methods

### 2.1. Participants and Procedure

One hundred and sixty parents (80 mothers and 80 fathers) of 80 Italian heterosexual families of toddlers aging 18 to 36 months (48 male) were recruited. Criteria for participation were full-term pregnancy, absence of hospitalizations lasting more than a week, and the absence of any physical/mental disability. The mean age for parents in our sample was 34 years (SD = 4.1) for males and 36 years (SD = 4.2) for females. Parents had high levels of education (for fathers, M = 18.2 years of education, SD = 1.8; and for mothers, M = 17.6, SD = 2.4), with only 3.1% of males and 1.8% of females reporting having completed high school or less.

Parents were recruited from kindergartens located in Rome, Italy. They received the questionnaires from trained psychologists who invited them to complete them independently at home. Parents were provided with a prepaid envelope which they used to return the questionnaires. Participants received written information on Italian privacy regulations and signed informed consent. Measures were completed at home. Data were gathered from September 2016 to March 2017. The study was approved by the Ethics Committee for Scientific Research of LUMSA University of Rome, Italy.

### 2.2. Instruments

#### 2.2.1. The Parent–Child Sleep Interaction Scale (PSIS)

The PSIS [1] is a 12-item parent-report questionnaire (items are rated on a 5-point Likert scale) that aims to assess bedtime behaviors and parent–child interactions related to problematic and dysfunctional sleep–wake patterns. We used the Italian version of PSIS that was adapted from English to Italian by a bilingual translator who back-translated the Italian version. The back-translation procedure from Italian to English proved to be identical in content with the original PSIS. The PSIS consists of three subscales: Sleep Reinforcement (SR; parental reassurance/reinforcement of child sleep behaviors; e.g., “I reassure my child that he/she is safe at night”; “I praise my child for good sleep behaviors”), Sleep Conflict (SC; conflict and child noncompliance surrounding sleep; e.g., “At bedtime, I remind/tell my child several times to go to sleep”; “I physically take my child to his/her room because of bedtime/sleep non-compliance”) and Sleep Dependence (SD; problems with independent sleep; e.g. “My child sleeps in my room all night”; “My child comes to my room at bedtime”). The "Total Score" index shows parental perception of hyper-involvement in children’s sleep routines, the presence of non-independent sleep patterns, and the need to be reassured and praised at bedtime.

#### 2.2.2. The Emotion Regulation Checklist (ERC)

The Emotion Regulation Checklist [8,9] is a 24-item parent-report measure of children’s self-regulation. We used the Italian adaptation by Molina et al. [9] that was validated with the Italian population. Items are rated on a 4-point Likert scale, and the checklist is composed of two subscales: the Lability/Negativity subscale (items representing a lack of flexibility, mood lability, and dysregulated negative affect) and the Emotion Regulation subscale (items describing situationally appropriate affective displays, empathy, and emotional self-awareness). The composite ERC score indicates the total emotion regulation level including both regulation and dysregulation.

#### 2.2.3. The Parenting Stress Index (PSI–Short Form)

The PSI [28,29] assesses the parental perception of stress perceived in relation to the child and parenting roles. We used the Italian adaptation by Guarino et al. [29] that was validated with the Italian population. Its 36 items are rated on a 6-point Likert scale, and three subscales can be distinguished, each consisting of 12 items: Parental Distress, Parent–Child Dysfunctional Interaction, and Difficult Child. Higher scores on the subscales and higher PSI-SF total scores indicated higher levels of perceived stress. The PSI shows excellent internal consistency and convergent validity with respect to prenatal stress, to other indices of postnatal stress, and to the quality of parent–infant interactions [28].

#### 2.2.4. The Social Provisions Scale (SPS)

This scale [17] consists of a 24-item self-report questionnaire and assesses individuals’ global perceptions of social support. We used the Italian version of SPS that was adapted from English to Italian by a bilingual translator who back-translated the Italian version. The back-translation procedure from Italian to English proved to be identical in content with the original SPS. Parents are asked to indicate on a 4-point Likert scale their agreement with items describing the social support available to them. Good test–retest reliability and convergent validity have been shown [17]. In the current study, two subscales were used: “Reliable Alliance” (the assurance that others can be counted upon for tangible assistance) and “Guidance” (ask for advice or information).

#### 2.2.5. Parental Involvement

Parental perceptions of their personal and their partner’s involvement in everyday and bedtime caring for children were evaluated by an ad-hoc questionnaire created by the authors comprising four single items, rated on a 5-point Likert scale (e.g., “How much do you take care of the child at bedtime?”; “How much does your partner take care of the child at bedtime?”). A combined score of perspectives from each mother and father was created. Higher scores indicated higher shared responsibility in children’s global and bedtime care.

#### 2.2.6. Sleep Quality Indicators

Sleep onset difficulties and nocturnal awakenings were evaluated with an ad-hoc questionnaire created by the authors and filled out by mothers. The sleep quality indicators were measured by two items indicating how many minutes were needed to settle down the child and how many times children wake during the night. Considering the Diagnostic Classification of Mental Health and Development Disorders of Infancy and Early Childhood [30] criteria to define Sleep Problems (Sleep Onset Disorder and Night Waking Disorder), we used sleep difficulties at bedtime (minutes that it takes the child to fall asleep) and the number of night wakings as children’s sleep quality indicators.

### 2.3. Data Analyses

The assumptions of the linearity and normality of the data were assessed and met; for this reason, we calculated the means and standard deviations for socio-demographic data and the Pearson correlation coefficient for bivariate correlations between the study variables. Multiple regressions were used to gain a further understanding of the relationships between the studied variables. Three separate multiple regression analyses were conducted by regressing maternal and paternal parenting stress and infant’s quality sleep in turn onto the correlated independent variables. The assumptions of the independence of observations, multicollinearity and homoscedasticity were assessed for each of the regression models and deemed to have been satisfactorily met. Analyses were performed with SPSS software version 24.0 (Armonk, NY: IBM Corp).

## 3. Results

One hundred and sixty parents (80 mothers and 80 fathers) of 80 Italian heterosexual families of toddlers aging 18 to 36 months (48 male) were recruited. Criteria for participation were full-term pregnancy, absence of hospitalizations lasting more than a week, and the absence of any physical/mental disability. The mean age for parents in our sample was 34 years (SD = 4.1) for males and 36 years (SD = 4.2) for females.

Bivariate Pearson correlations among the key variables under study are presented in Table 1. Correlations demonstrate suitably moderate associations among the variables included in the regression models, suggesting acceptable amounts of collinearity for the planned multivariate analyses. Variables taken into account for the regression models were the minutes needed to settle down the child, number of nocturnal awakenings, emotion regulation, bedtime routines, parental bedtime involvement, parental perceived social support, and stress; these were included in a new model with the aim of explaining toddlers’ sleep quality and parental stress.

Three separate multiple regression analyses were conducted to investigate the contributions of the relevant, correlated independent variables to the variance of the dimensions of sleep quality. A linear multiple regression was performed in order to assess the contributions of each variable on maternal and paternal parental stress, respectively. In line with the reviewed literature, quality sleep variables, infant’s emotion regulation dimensions and parental psychosocial functioning variables were entered into the model. The results of the regression analyses are presented in Table 2, Table 3 and Table 4. The first linear regression (F_(9,49)_ = 4.48; *p* < 0.05) revealed that the entered variables explained 50% of the variance of paternal parenting stress. In the model, paternal perceived social support was found to be the strongest predictor (*β* = −0.33, *p* = 0.008). The other predictors were the number of minutes required to settle down the children (*β* = 0.26, *p* < 0.05) and paternal perception of the child’s emotional lability (*β* = 0.29, *p* < 0.05).

Regarding maternal parenting stress, the regression model explains 57% of the variance of maternal parenting stress (F_(9, 43)_ = 6.47, *p* < 0.001). In the model, maternal social support (*β* = −0.43, *p* < 0.001) and paternal global involvement in caring (*β* = −0.24, *p* = 0.051) negatively predicted maternal parenting stress, whereas maternal perception of their child’s emotional lability (*β* = 0.37, *p* = 0.002) and number of nocturnal awakenings (*β* = 0.30, *p* = 0.006) were significant positive predictors of maternal parenting stress.

Finally, a linear regression was used to assess the independent variables’ contribution to the variance of the number of night awakenings (F_(9, 43)_ = 7.57, *p* < 0.001). The model explains 36% of the variance of the number of night awakenings reported by mothers. Maternal parenting stress was a significant positive predictor (*β* = 0.35, *p* < 0.05), whereas paternal global involvement in caring negatively predicted (*β* = −0.33, *p* < 0.05) the number of night awakenings.

## 4. Discussion

The main results of the current study confirmed our first hypothesis (H1). Firstly, our data suggested that maternal and paternal stress are linked to toddlers’ quality of sleep: the number of night awakenings reported by mothers and the time required by their children to fall asleep were significantly associated with parental distress. These outcomes offer evidence for the potential agency of toddlers within the family system [31]. Sinai and Tikotzky [32] suggested that parental perceptions of children’s sleep behaviors may significantly impact parenting stress: parents who considered their toddlers’ sleep as challenging reported higher levels of stress. Furthermore, the results of the regression analysis suggested that maternal and paternal stress is related to toddlers’ emotional lability. The tendency to express negative emotionality at lower thresholds with higher intensity may induce children to rely more strongly on caregivers to obtain comfort, probably eliciting an inconsistent or less sensitive parenting response [6].

Our second hypothesis (H2) was partially confirmed. Our results show that social support may act as a protective factor for parental stress. Parents who feel supported by their social context show greater emotional availability towards their children, fostering toddlers’ emotion regulation abilities [2]. Researchers have typically focused little on paternal contributions to infant care [19]. According to the existing literature, we found that paternal involvement in children’s everyday care as perceived by mothers negatively predicted the mothers’ levels of parenting stress. A balance in caregiving responsibilities and stable cooperation between parents can reduce maternal stress, positively influencing maternal and children’s sleep [23]. These results may suggest a potential moderation effect of parental involvement in children’s care on the associations between children’s emotion regulation and parental stress, which should be investigated in more depth in future research.

Overall, we found that higher parental functioning is related to fewer nocturnal awakenings. Our results partially confirmed the third hypothesis (H3), showing maternal stress to be a risk factor for increasing night awakenings. However, we confirmed the fourth hypothesis (H4), highlighting paternal involvement in children’s bedtime care as a protective factor for sleep quality [24,33]. Our data confirm the results of previous studies showing associations between higher levels of maternal stress and bedtime difficulties, less favorable child sleep profiles, and challenging bedtime routines [15,16]. Regarding parental involvement, several studies identified a relationship between paternal factors and children’s sleep patterns [19]. Millikovsky and colleagues [34] found paternal involvement to moderately influence children’s sleep, providing emotional and instrumental support for mothers and reducing their stress levels.

The results of the current study shed light on the specific workings of reciprocity within the family system. Greater perceived social support and paternal involvement potentially enhance the parental dyad’s capacity to jointly address a child’s sleep problems.

This study has several limitations that should be pointed out. First of all, the sample size precluded us from conducting more sophisticated analyses that better modelled the transactional nature of the associations of interest. Moreover, the regression model showed only relations between constructs and not causality. In future research, longitudinal studies are needed. Finally, an objective sleep measure such as actigraphy or polysomnography could provide a more reliable assessment of sleep quality.

## 5. Conclusions

During the first years of life, the development of the adequate regulation of sleep–wake patterns represents a challenging process for toddlers as well for the whole family.

The results from this study show that dimensions related to the proximal familial context can play a role in shaping bedtime habits and practices. In particular, our results emphasized the protective role of parental perceived social support for parenting stress and paternal bedtime involvement for children’s sleep quality. Couples who share caregiving responsibilities are characterized by higher levels of perceived support and lower levels of stress, promoting the regulation and consolidation of their children’s sleep patterns, in terms of the time needed to fall asleep and the number of night-time awakenings. The outcomes of this study confirm the complex and composite nature of sleep in toddlerhood, approaching the transactional family dynamics framework [25] and shedding light on the considerable role of familial context and wellbeing in infant sleep.

The results highlight some relevant practical implications that need to be taken into account. Healthcare professionals have a considerable role to play, not only in guiding families in the early recognition of sleep disturbances but also in planning and realizing tailored models of intervention to be applied as promptly as possible in risk conditions. Interventions should be focused on supporting families and toddlers in their attempts to achieve adequate sleep patterns and would consequently benefit children’s cognition, behavior, and affects, which are strongly dependent on sleep.

Moreover, other specialized figures involved in toddlers’ child-care can surely benefit from our research conclusions; school professionals may take advantage of the literature recommendations concerning sleep not only in their everyday practical work in early infancy educational contexts but also in terms of focusing on and sharing effective sleep management strategies with parents, indirectly supporting toddlers’ child-care.

Further research is necessary and may contribute to deepening our understanding and raising awareness of the relationships between proximal context and toddlers’ sleep, identifying risk and protective factors in children’s sleep patterns. 

## Figures and Tables

**Table 1 ijerph-17-02494-t001:** Correlations among variables under study.

Variables	2	3	4	5	6	7	8	9	10	11	12	13
1. Children’s Age	0.23	0.11	0.05	0.26 *	0.04	0.05	0.00	−0.02	−0.39 **	−0.27 *	−0.01	−0.08
2. Min. settle down	1	−0.14	0.01	−0.05	0.33 *	0.37 **	0.03	−0.18	0.28 *	0.25	0.20	0.16
3. N. Night awakenings		1	−0.27 *	−0.04	0.14	−0.05	−0.31 **	0.06	−0.07	−0.01	0.05	−0.01
4. Paternal bedtime involvement (P)			1	0.33 *	−0.18	0.14	−0.01	−0.25	−0.13	−0.17	−0.23	−0.35 **
5. Paternal global involvement (P)				1	−0.12	0.04	−0.13	−0.09	0.06	−0.07	0.00	−0.27 *
6. Paternal problematic bedtime routines					1	0.71 **	−0.05	−0.03	0.20	0.12	0.38 **	0.36 **
7. Maternal problematic bedtime routines						1	0.59	0.06	0.08	0.11	0.34 **	0.47 **
8. Paternal perceived support							1	0.29 *	−0.14	0.02	−0.04	−0.03
9. Maternal perceived support								1	−0.10	−0.22	0.02	0.02
10. Paternal parenting stress									1	0.79 **	0.21	0.05
11. Maternal parenting stress										1	0.15	0.15
12. Paternal emotional lability											1	0.62 **
13. Maternal emotional lability												1

***p* < 0.01; **p* < 0.053.1 multivariate analyses.

**Table 2 ijerph-17-02494-t002:** Regression coefficients of predictors of paternal parenting stress.

Variables	Paternal Parenting Stress			
	B	E.S.	β	p
Min. settle down	0.18	0.09	0.26	0.045
N. Night awakenings	2.24	2.56	0.11	0.386
Paternal perception of child’s emotional lability	1.13	0.52	0.29	0.036
Paternal perceived support	−1.47	0.52	−0.33	0.008
Paternal problematic bedtime routines	0.35	0.27	0.18	0.203
Paternal global involvement (P)	1.42	2.77	0.06	0.611
Paternal bedtime involvement (P)	0.84	2.38	0.05	0.726
Maternal global involvement (P)	−2.23	3.37	−0.10	0.511
Maternal bedtime involvement (P)	3.99	2.65	0.23	0.14
R = 0.70 R^2^ = 0.50				

*Note.* Min. = minutes; N. = number; P = reported by fathers.

**Table 3 ijerph-17-02494-t003:** Regression coefficients of predictors of maternal parenting stress.

Variables	Maternal Parenting Stress			
	B	E.S.	β	p
Min. settle down	0.09	0.91	0.11	0.318
N. Night awakenings	6.09	2.12	0.30	0.006
Maternal perception of child’s emotional lability	1.49	0.46	0.37	0.002
Maternal perceived support	−2.09	0.50	−0.43	0.000
Maternal problematic bedtime routines	−0.29	0.24	−0.14	0.221
Maternal global involvement (M)	−2.78	2.63	−0.12	0.296
Paternal global involvement (M)	−4,66	2.52	−0.24	0.051
Maternal bedtime involvement (M)	−0.66	2.50	−0.03	0.791
Paternal bedtime involvement (M)	−0.55	2.01	−0.03	0.786
R = 0.75 R^2^ = 0.57				

*Note.* Min. = minutes; N. = number; M = reported by mothers.

**Table 4 ijerph-17-02494-t004:** Regression coefficients of predictors of the number of night awakenings.

Variables	N. Night Awakenings			
	B	E.S.	β	p
Children’s Age	−0.01	0.02	−0.09	0.518
Min. settle down	−0.01	0.01	−0.19	0.249
Paternal emotional lability	0.01	0.04	0.05	0.833
Maternal emotional lability	−0.04	0.04	−0.19	0.394
Parental perceived support	−0.05	0.04	−0.24	0.146
Paternal problematic bedtime routines	0.03	0.02	0.36	0.144
Maternal problematic bedtime routines	−0.02	0.02	−0.15	0.514
Paternal global parenting stress	−0.01	0.01	−0.18	0.438
Maternal global parenting stress	0.01	0.01	0.35	0.050
Paternal bedtime involvement	−0.28	0.14	−0.33	0.046
Maternal bedtime involvement	−0.06	0.14	−0.08	0.638
R = 0.60				
R^2^ = 0.36				

*Note.* Min. = minutes; N. = number.

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
