# Peer review of "Predictive Factors of Toddlers’ Sleep and Parental Stress"

_ijerph, 2020, doi:10.3390/ijerph17072494_

Round 1

Reviewer 1 Report

  1. Abstract: description of the methods used is needed in the abstract, in terms of the tool used and the study design.
  2. Main concern is the description of the objectives of the study: authors keep mentioning “to explore”, which gives an indication about a qualitative approach. I think they should use: to investigate, to evaluate instead.
  3. The introduction is well-written, but there is no need to the sub-sections. Better if the authors keep the flow of information without cutting them into sub-sections.
  4. The results section, and especially the tables need some improvement, in terms of clarity and presentation. Example is table.1. where the design and the content are not clear to the reader.

Author Response

1.Abstract: description of the methods used is needed in the abstract, in terms of the tool used and the study design.

Answer: following your suggestions tool used and study design were added (line 16-21)

  1. Main concern is the description of the objectives of the study: authors keep mentioning “to explore”, which gives an indication about a qualitative approach. I think they should use: to investigate, to evaluate instead.

Answer: following your suggestions the term “to explore” was replaced throughout the text

3.The introduction is well-written, but there is no need to the sub-sections. Better if the authors keep the flow of information without cutting them into sub-sections.

Answer: we eliminated the sub-sections

4.The results section, and especially the tables need some improvement, in terms of clarity and presentation. Example is table.1. where the design and the content are not clear to the reader

Answer: we tried to better explain your concerns, please see the new version of the manuscript

Reviewer 2 Report

Although the topic of this manuscript addresses is a very interesting for the special issue "Child and Adolescent Health in a Life-Course Perspective", there is a lack of understanding and clarity on message of the manuscript that may limit the interest for the readers. Moreover, there are some concerns that should be clarified as well as extra information should be included to permit the reproducibility of the study by other researchers.

  1. The TITLE should be rewritten. It does not clearly reflect the objective or issue of the

  1. In ABSTRACT, it is necessary that:
    • Background includes the aim of the
    • Methods section should be including information about how the study was performed (participants, statistical process, material).
    • Results section should include quantitative results of the aim of the
    • Conclusion should answer the

  1. The AIM of the study is not clear. Are there two different objectives?
    • The first one is to exam the correlations between all this variables: toddlers’ sleep quality, emotion regulation, bedtime routines, parental bedtime involvement, parental perceived social support and stress?
    • And the second one is to study the relation only between toddlers’ sleep quality and parental stress?

The authors should present the objective or objectives clearer, as well as the hypotheses.

  1. In MATERIALS AND METHODS:
    • Sociodemographic characteristics present between lines 112 and 114 should be included in the
    • The “Mmales” and “Mfemales” expressions should be replaced by Mean (SD, i.e. standard deviation).
    • In point 1 Participants and procedure, it is necessary to add more information about the precedence of participants, when and where was the study carried out, who carried out the evaluations….)
    • Authors don’t indicate the version of the instruments used (the original or Italian version).
    • Are all the instrument used validated for Italian population?
    • Why authors include examples of some items from different instruments?
    • In point 2.3 Data Analyses:
      • Authors don’t refer the software used for statistical
      • It is not indicated if the authors have checked the normality of the data to know if data should be presented as means or medians, or use Pearson or Spearman correlation.

  1. In RESULTS:

  • At the beginning of results, the analyses of study should be explicitly
  • The authors should include information about the sociodemographic and socioeconomic variables included in the
  • In line 175, about the text "the key variables”: how these variables were determined?
  • In line 176, what are the variables refer in the text?
  • In table 1:
    • The title should be rewritten because it does not reflect the results that are presented.
    • It’s is necessary indicate as footnote the meaning of numbers in row
    • Pearson’s correlations coefficient (r) need to be specified, not only the p-value.
  • In point 3.1, the three separate multiple regression analyses don’t answer the aim of the study. The response or dependent variable should be toddlers’ sleep
  • In tables 2.4 it is necessary indicate as footnote the meaning of individual letters, initials and

  1. In the first paragraph of DISCUSSION, the authors should summarize of the most important results and not repeat the objective of

  1. In CONCLUSIONS:
    • According to instructions for authors of the journal, this section should provide a brief summary of the main conclusions, so the text between line 252 and 257 should be removed.

  1. Review For example, reference 1 (line 291), the journal is not indicated.

Author Response

1.The TITLE should be rewritten. It does not clearly reflect the objective or issue of the study

Answer: Following your suggestion the title has been modified

  1. In ABSTRACT, it is necessary that: Background includes the aim of the study

Answer :  Following your suggestion we have added

Methods section should be including information about how the study was performed (participants, statistical process, material).

Answer: Following your suggestion we have added

Results section should include quantitative results of the aim of the study

Answer: Although we agree with you we couldn’t add since we must not exceed 200 words

Conclusion should answer the…

Answer :  Following your suggestion we have added

The AIM of the study is not clear. Are there two different objectives?

The first one is to exam the correlations between all this variables: toddlers’ sleep quality, emotion regulation, bedtime routines, parental bedtime involvement, parental perceived social support and stress?

And the second one is to study the relation only between toddlers’ sleep quality and parental stress?

The authors should present the objective or objectives clearer, as well as the hypotheses.

Answer : We clarified the aim of the study and our hypotheses

In MATERIALS AND METHODS:

Sociodemographic characteristics present between lines 112 and 114 should be included in the section of results

Answer : Sociodemographic characteristics had been added in the Result Section.

The “Mmales” and “Mfemales” expressions should be replaced by Mean (SD, i.e. standard deviation).

Answer : The expressions had been replaced as you have suggested

In point 1 Participants and procedure, it is necessary to add more information about the precedence of participants, when and where was the study carried out, who carried out the evaluations….)

Answer : Information were added following your suggestion

Authors don’t indicate the version of the instruments used (the original or Italian version).

Are all the instrument used validated for Italian population?

Answer : Information were added following your suggestion

Why authors include examples of some items from different instruments?

Answer : We have decided to include items for the instruments not very well-known

In point 2.3 Data Analyses:

Authors don’t refer the software used for statistical

Answer : Informations were added following your suggestion

It is not indicated if the authors have checked the normality of the data to know if data should be presented as means or medians, or use Pearson or Spearman correlation.

Answer : Informations were in Data analyses

In RESULTS

At the beginning of results, the analyses of study should be explicitly

The authors should include information about the sociodemographic and socioeconomic variables included in the

Answer : Sociodemographic data were included in the Result section

In line 175, about the text "the key variables”: how these variables were determined?

In line 176, what are the variables refer in the text?

Answer : Variables had been specified at line 193

In table 1: The title should be rewritten because it does not reflect the results that are presented.

It’s is necessary indicate as footnote the meaning of numbers in row

Pearson’s correlations coefficient (r) need to be specified, not only the p-value.

Answer : Informations were added following your suggestion

In point 3.1, the three separate multiple regression analyses don’t answer the aim of the study. The response or dependent variable should be toddlers’ sleep

Aim of the study was better specified in the Introduction section

In tables 2.4 it is necessary indicate as footnote the meaning of individual letters, initials and

Answer : Informations were added following your suggestion

In the first paragraph of DISCUSSION, the authors should summarize of the most important results and not repeat the objective of

Answer : The text had been modified accordingly

In CONCLUSIONS:

According to instructions for authors of the journal, this section should provide a brief summary of the main conclusions, so the text between line 252 and 257 should be removed.

Answer : The text had been modified accordingly

Review For example, reference 1 (line 291), the journal is not indicated.

Answer : All references had been reviewed

Round 2

Reviewer 2 Report

Thank you for the effort to improve the quality of the paper. However, in line 94, the word “stress” should be deleted.